# The Effect of Social Recognition on Support for Climate Change Mitigation Measures

**Stephanie Jütersonke * and Martin Groß**

Department of Sociology, University of Tübingen, 72074 Tübingen, Germany; martin.gross@uni-tuebingen.de
* Correspondence: stephanie.juetersonke@uni-tuebingen.de

**Abstract:** Social recognition is introduced as an explanatory factor for support for climate change mitigation measures to complement already existing research. Drawing on social identity theory, it is established that respect emanating from being part of a generation and social class increases support for climate policies through positive influence on self-assessed financial situation, trust in political institutions, generalized trust, and solidarity. Considering the costs and benefits of climate policies, it is assumed that the importance of respect varies between generations and social classes. Analyses are based on data which were collected via online survey (n = 3046) in September and October 2022 that are representative of the German resident population. Results from linear regressions and structural equation modeling corroborate that the influence of social recognition varies between generations and that it operates through the suggested mechanisms. The most important of these mechanisms is the strengthening of solidarity and trust in political institutions by social recognition.

**Keywords:** support for climate policies; climate change; collective good problem; social recognition; trust in political institutions; solidarity; social identity theory

## 1. Introduction

Leading institutions in climate research are urging political leaders for quick action to prevent the worst consequences of global warming [1]. Yet, every option for effective action hinges on whether the public supports it and some attempts to implement such measures have been met with angry resistance in the past (e.g., [2–5]). The question of how support for climate policies could be strengthened is therefore of utmost importance for climate protection and has been at the center of many current and earlier research articles.

Previous research has looked at the topic from numerous angles by assessing the impact of socio-demographic features, values, or political attitudes on support for climate protection (e.g., [4–9]).

Our approach complements this body of research by suggesting social recognition as a new explanatory factor which may not only have the potential to predict support for climate change mitigation measures but could also explain *how* previously discussed influence factors such as institutional trust (e.g., [10]) affect this support. The central argument of this paper puts the human need for recognition at its core and combines ideas from social psychology [11,12], political science [13], sociology, and philosophy [14]. While a general need for recognition, or respect, on the individual level is well established [15,16], we emphasize the importance of group belonging, and the respect people receive for being part of social groups for building self-esteem and forming political attitudes. This focus on social groups and what is called *social* recognition is particularly promising for the study of pro-environmental attitudes because climate protection is a collective good problem (e.g., [17–19]) that largely depends on coordination by superordinate institutions [20] which often interact with individuals by referring to them as members of social categories [21].

In developing our argument, the article proceeds by elaborating on climate protection as a collective good and pointing out the specific difficulties that arise from the varying

degrees to which generations and social classes are affected by climate change and climate policies. In drawing on social identity theory, social recognition for generations and social classes are introduced as dimensions of respect that are highly salient in the public discourse on climate protection. We argue that feeling (in)sufficiently recognized for being part of a generation as well as social class is an important influence factor on self-assessed economic situation, solidarity, generalized trust, and trust in political institutions and that all of these have a significant influence on support for climate protection measures. After concluding the theoretical section with our hypotheses, we present results from linear regression analyses as well as structural equation modeling that are based on data collected in an online survey conducted in September and October 2022. This paper concludes with a general discussion of our results and suggestions for further research.

## 2. Theoretical Considerations

To first specify the problem, the following paragraph elaborates on the collective good nature of climate change and the added difficulty that arises because different social groups are affected differently by climate change as well as climate change mitigation measures. To better understand these difficulties, the second paragraph describes how differences within social classes and generations come about. The third paragraph elaborates on what might solve the challenges that stand in the way of collective action and the fourth introduces social recognition as an important dimension of respect that has undergone tremendous changes over the last decades which has been shown to affect social cohesion. In combining these thoughts, the fifth paragraph highlights the salience and importance of social recognition in the climate change discourse and presents four mechanisms which link social recognition and pro-environmental attitudes. Finally, the final section concludes with the presentation of our hypotheses.

### 2.1. Climate Change Mitigation as a Critical Collective Good Situation

Climate protection is a classic collective good problem in the sense that the outcome (1) is desirable for everyone, (2) requires the contribution of as many people as possible, and (3) cannot be protected from free riders [22,23]. The situation that arises can be described as a 'critical collective good situation' [19], (p. 42) or 'social dilemma' [18], in which every individual is better off by not cooperating, yet the individual pay-off decreases if everybody refuses to cooperate [24]. In this setting, self-interested individuals have little reason to act in a way that is beneficial for the collective good because others might not contribute yet reap the rewards of their own efforts. Motivation is further dampened because individual contributions only have a marginal influence on the provision of the collective good [24,25]. In case of the global climate 'the individual contribution . . . is usually so insignificant and often the object of such a low social control that it is not covered by sufficient incentives' [19], (p. 48). For achieving the collective good, people therefore must make sacrifices that run counter to their immediate rational interests. These sacrifices can vary in strength, depending on the extent to which the collective good is of equal interest to each member of a group, with weak sacrifices relating to cases in which all members of a group can benefit equally and strong sacrifices relating to situations in which the collective good is of less benefit to some [19] (p. 49). This aspect further complicates cooperative action against global warming because different social groups are differently affected by the risks of global warming and the implications of climate policies.

### 2.2. Differences between Social Groups Regarding Climate Change Mitigation

If large enough and not cushioned by any mediating influence factors, the differences in risks and benefits of climate change and climate policies can result in an overly divergent incentive structure that has the potential to undermine any motivation for collective effort.

First, the consequences of climate change and benefits of climate protection will affect people of different generations differently. A continued rise in global temperature poses a severe threat to human health [26–28] which will on average affect the lives of younger

generations for a longer time and to a greater degree than those of older generations. For this reason, generational belonging has already been introduced as a potential influence factor by previous research [9,29,30]. In addition, climate change mitigation measures imply increasing costs for many products as well as other constraints on consumption and behavior. These implications have a different meaning for people of different generations. While the higher risks of climate change make these costs an investment in one's future for younger people and thus a weak sacrifice, these costs are a strong sacrifice for people of older generations because for them, who will on average be less affected by the consequences of global warming, they are expensive in the present and of uncertain value in the future [19,31].

Second, different social classes are also likely to have different attitudes towards the costs of climate protection. The topic of social class belonging is highly salient in the climate discourse because on the one hand, the conversation often centers around class-specific consumption and lifestyles. On the other hand, price increases are a proportionally greater burden for people with smaller incomes [32]. Adding to that, climate change mitigation will require a restructuring of the economy and workplace which will affect occupational groups differently [33,34] and which some will find easier to adapt to than others.

### 2.3. Factors That Promote Support for Climate Change Mitigation Measures

Against the background of the collective good structure of climate protection, and the different distribution of risk and burden between social categories, we can assume that cooperative action against global warming cannot be sustained by rational individual interest alone but must be mediated by additional influence factors [19,35].

First, there must be some trust in others and in their willingness to cooperate for the collective good, because making sacrifices puts the individual in a vulnerable position and only makes sense if a critical number of others do the same. Therefore, people must have some level of *general trust* which justifies the assumption that others will, on the one hand, contribute their part [4,18,36] and, on the other hand, offer support to the individual in times of vulnerability and need [37,38]. Second, people need to have *trust in the institutions* that coordinate collective action and believe that the contributions that political actors ask for will be used effectively and in the common interest. To support climate protection, they need to expect political institutions to be willing and able to act in the sense of the collective good [39]. Third, people need to be confident that increases in *costs will not pose a fundamental threat* to their livelihood. While individual resources clearly play an important role here, we assume that the subjective perception that society and political institutions would support the individual in times of need also contributes to their willingness to support climate policies.

Finally, there needs to be a *commitment to solidarity norms* that discourages free riding. Even in cases in which people see measures as useful and want to benefit from the collective good, they might also hope that the contributions of others will be sufficient without them doing their part. A commitment to solidarity norms can prevent such behavior [19].

Solidarity is particularly important in climate protection because people, and especially generations, are affected differently by global warming. For older generations, the support for climate protection must be rooted in the consideration of other people's interests.

It is the argument of this paper that social recognition plays a central part in strengthening pro-environmental attitudes by positively influencing all of these mediators.

### 2.4. Social Recognition

The first theoretical thinkers on recognition refer to recognition as a feeling which individuals experience in intersubjective relations with relevant others (e.g., [40,41]). The need for recognition, or respect, has been identified to be one of the fundamental human motives and as essential for developing self-esteem and self-confidence [15]. To reach full-fledged personhood and complete integration into society, individuals must feel seen and recognized for their achievements as well as for their own sake by interaction partners and institutions [40].

Approaches centering on social recognition connect these ideas relating to individual needs with Social Identity Theory (SIT), a line of thought that was developed by Henri Tajfel and John C. Turner [11]. SIT states that people's self-concept or identity is influenced by their membership in various social groups [42] (p. 69). Whether this influence is positive or negative depends on whether people perceive their group to be superior or inferior to outgroups. Self-esteem is therefore largely based on a favorable comparison between in-groups and out-groups, which makes these comparisons highly important for the individual [12].

Early experimental SIT research shows that feelings of group belonging and consequent ingroup favoritism and outgroup discrimination can easily be evoked by assigning group membership through arbitrary means [43]. When applying the reasoning of SIT to the study of social groups which are less arbitrary and easy to change such as ethnicities, nationalities, genders, or age groups, similar patterns emerge [44]. These social groups which are rooted in relatively more solidified and/or visible differences and connect people who are not necessarily in direct interaction with another are referred to as social categories. Perceiving the world in terms of social categories influences the perception of self and others and shapes interactions between individuals and groups [45]. As social categories are nothing but more tangible and, in some cases, institutionalized kinds of social groups, processes of comparison between the ingroup and outgroup unfold with the same consequences for individual self-conception and self-esteem. There is a two-fold difference, however, which relates to the degree to which comparisons are extended and institutionalized in society.

First, social categories often comprise much larger groups and therefore encourage comparison with not only people and groups on an interpersonal but also on a more abstract, societal level.

Second, macro-level institutions such as governing bodies often relate to individuals as members of social categories in both rhetoric and action by, for example, highlighting the needs of some groups and criticizing others, or by implementing measures of affirmative action [21]. In selecting the focus of their attention and criticism and in implementing measures that support the interests and needs of social categories unevenly, interactions between institutions and people communicate benevolence or critique, concern or neglect towards social categories. As part of public discourse, this different treatment of social categories becomes highly salient for people in everyday lives and can inspire the perception that institutions recognize people of different social categories differently. This perception further encourages and influences the tendency of people to compare their social category with people of other social categories, which translates into positive or negative contributions to self-esteem.

Even though the evaluations that assign differing values of recognition to social groups tend to be socially consensual [11] and are hard to change once they have become institutionalized, they can be contested. Yet, the crucial importance of social recognition makes people react sensitively to changes in the recognition order when they expect their superior position to be challenged [13].

The need for recognition of social categories has been at the center of the public debate for some time now, as processes of macro-social change such as globalization, liberalization, and modernization have caused major shifts in the recognition order, with consequences for social life that have yet to be fully understood. In analyzing the consequences of these social change processes Francis Fukuyama [13] identifies perceived losses in recognition by traditionally advantaged groups, while previously marginalized groups gain more recognition as one important reason for polarization and weakening of social cohesion in the U.S. For Germany, studies as well find a relation between perceived lack of social recognition and attitudes that threaten social cohesion [46,47].

These findings are highly relevant for the analysis of pro-environmental attitudes because social cohesion is an important prerequisite for the support of climate change mitigation policies [48] in a relationship that is rooted in the collective good structure of climate protection.

In combining the thoughts of all previous paragraphs, the next subsection will elaborate on social recognition and the influence it has on climate change mitigation measures.

*2.5. Linking Social Recognition and Pro-Environmental Attitudes*

At its core, all mechanisms relating social recognition to pro-environmental attitudes understand recognition as a signal which tells the individual that their group is an important part of society whose contributions matter [11], or put differently, perceiving social recognition is understood as feeling respected [13]. Issues of respect are highly present in the climate discourse and particularly salient when it comes to the social groups that are differently affected by climate change as well as climate policies, which especially relates to generations and social classes. In both cases, the feeling of getting an (in)appropriate amount of respect for the specific group is subject to public discussion and in both cases the difference in affectedness as well as the way that the discourse is led has the potential to divide people and undermine cooperation.

Relating to generational belonging, younger and older generations alike accuse the other of being disrespectful. While older generations see the demands for stricter regulation and the restriction of consumption as excessive and threatening to the standard of living that they consider to be the well-deserved reward for their life's work, younger generations accuse them of not taking their fears seriously and destroying their future livelihood [49].

Relating to social class belonging, the discourse on climate change is inclined to alienate the working and lower classes. First, the emphasis that the middle class in western countries puts on what is called ethical consumption results in a debate which implicitly puts the blame for global warming on those who do not participate in said ethical yet expensive lifestyle [50]. Second, per capita $CO_2$ emissions increase with income, which in the German case means that the richest 10 percent of the population emit almost as much $CO_2$ as the poorest 50 percent [51]. It can be assumed that the implicit allegations for not acting and consuming the "right" way as well as knowledge about the high $CO_2$ emissions of the upper classes have a stifling effect on motivation to support climate change mitigation measures for the lower and working classes.

These differing dynamics within either the social class or generation category, which relate to the key differences that characterize each category, suggest that our analysis needs to address specific dimensions of respect for both of them.

First, belonging to a social class is closely connected to occupation. It is therefore reasonable to assume that feeling respected for being part of a social class is likely to be reflected in the perception that society and politics appreciate the work of one's group and perceive classes to be of equal value for society. Second, being part of a generation connects people who are in similar phases of their life and who, related to that, share some common interests. It is therefore likely that the feeling of receiving respect for one's generation is reflected by the perception that society and politics care about these generation-specific interests which are often expressed in terms of needs and fears.

Building on all previous considerations, four mechanisms are suggested through which social recognition strengthens pro-environmental attitudes among people of different social classes and generations, with the first and second mechanism affecting all people equally, and the third and fourth mechanism affecting people differently depending on the social class or generation to which they belong.

First, feeling respected by society and political agents conveys that others are benevolent, and benevolence is one of the main dimensions of trustworthiness [52]. Perceiving others as benevolent therefore encourages those who are recognized to put their trust in others, and generalized trust is an important prerequisite of contributing to the collective good [4,18,36].

Second, feelings of respect and benevolence are also important for fostering political trust, which again is an important prerequisite for approval of climate change mitigation measures [10, 53,54].

The third mechanism relates to the fact that feeling respected by society and politics is likely to have a strong influence on how people assess their social status or position

within the social structure of society. People who feel that their group is respected consider themselves to be an important part of society and are confident that they would receive support when in need. This implies that social recognition has the potential to cushion fear of costs of climate change mitigation. These aspects should be particularly important for people that belong to lower classes and the working class because feeling respected should be able to mitigate economic fears for those who are particularly affected by the costs of climate change policies. It should also be able to remedy feelings of alienation caused by implicit accusations against their lifestyle and the extreme inequalities in $CO_2$ emissions.

Finally, the prospect of experiencing solidarity that is fostered by recognition will cause the individual to respect the interests of others because it activates norms of reciprocity. In cases of support for climate policies this implies that people are willing to accept costs and inconveniences for the collective good even if they are unsure about their own benefits. Relating to generation specific differences, a strengthening of solidarity through recognition should be relevant especially for older people, as these costs are a weak sacrifice for younger people which they should accept out of self-interest.

*2.6. Hypotheses*

Building on these thoughts and arguments we present the following hypotheses.

**Hypothesis 1.** *Recognition (class recognition as well as generational recognition) has a positive influence on pro-environmental attitudes.*

While this hypothesis addresses the overall effect of recognition, hypotheses 2 and 3 address differences in benefits and consequences of climate policies within categories. First, regarding the benefits of climate protection and the assumed influence of social recognition on solidarity, we state:

**Hypothesis 2.** *The assumed influence of recognition is more important for people of older generations.*

Second, with respect to the exclusionary discourse on climate protection and the costs of measures that will place a disproportionate burden on the lower and working classes we state that:

**Hypothesis 3.** *The assumed influence of recognition is more important for people of the lower and working classes.*

In addition to these direct effects of social recognition on climate protection, the previous section established that the effects should also operate through additional influence factors.

First, social recognition was suggested to be an important indicator for benevolence of institutions and others, which encourages the trust in political entities and other individuals that is necessary for people to support pro-environmental attitudes (e.g., [4,10]).

**Hypothesis 4.** *Social recognition has a positive influence on pro-environmental attitudes by strengthening institutional trust.*

**Hypothesis 5.** *Social recognition has a positive influence on pro-environmental attitudes by strengthening general trust.*

Second, it was assumed that social recognition can encourage a positive assessment of one's financial situation and therefore cushion the fear of financial burden by communicating to the individual that their social group is important for society:

**Hypothesis 6.** *Social recognition encourages a positive assessment of one's financial situation, thus strengthening support for climate protection measures.*

Lastly, we suggested that social recognition can increase the solidarity that is necessary for combating climate change [31,55] because it makes people anticipate solidarity in other circumstances.

**Hypothesis 7.** *Social recognition has a positive influence on pro-environmental attitudes by increasing solidarity.*

## 3. Data, Methods, Variables and Models

This section provides information on data collection, methodological strategy, and variables.

### 3.1. Data

The data were collected in September and October 2022 through an online survey conducted by the access panel provider "Kantar". "Kantar" was also in charge of participant recruitment and drew respondents from the participant pool of the "Payback Panel". This panel only relies on active offline recruitment, thereby preventing self-selection. A total of 3124 of the 7500 participants that were invited completed the questionnaire, which results in a response rate of 41.1 percent. After checking for dubious response behavior (e.g., people with a response time far below (7.6 min) or above (more than 70 min) the median response time and respondents with more than 10% of item non-response) all in all 78 respondents had to be excluded, which left us with a final sample of 3046 respondents representing the adult population of Germany by virtue of age, sex, and geographic provenance (for variable distributions please see Appendix A).

The questionnaire was constructed as a planned missing design that consists of core questions which all participants answer and three blocks of additional questions that are split between participants. Median response time for the questionnaire was 19.1 min.

### 3.2. Methods

The methodological strategy of the paper comprises two main approaches: regression analyses and structural equation modeling (SEM). Linear regression models are used to provide insights into the general importance of social recognition, which is stated in hypothesis 1, and allow assessments of differences in the importance of recognition by social categories as stated in hypotheses 2 and 3. They can also give first insights regarding the importance of mediators that is stated in hypotheses 4 to 7. Missing values were imputed using multiple imputation [56] via the R package "mice" (version number: 3.14.0) [57].

Structural equation modeling (SEM) is used to assess the mediating role of self-assessed financial situation, trust, and solidarity that is stated in hypotheses 4 to 7. For SEM, a Full Information Maximum Likelihood (FIML) estimation algorithm is used to handle missing data (R package "lavaan"; version number: 0.6.15) [58].

### 3.3. Variables

For our measurement instruments we relied on established measures where possible and developed new scales where necessary. All scales were subject to extensive pre-testing before the main study (please look at Appendix A for factor loadings in Table A1).

The dependent variable is a self-developed construct of three items which measure (dis)agreement with a particular climate protection measure on a 5-point Likert scale. Using various items to assess support for measures is advantageous because it allows us to capture attitudes towards the concept "climate protection" rather than (dis)approval for single measures. The first and second items both refer to increases in food prices, one through higher meat taxation and the other through cuts in subsidies for non-organic agriculture. The third item measures support for banning of single-use packaging with the consequence of less convenient take-away options. We think that these items are especially suited to measure support for climate protection among people of all social classes and generations because they relate to everyday consumption, which has been identified as one of the main

drivers of global warming [59]. Adding to that, they are less affected by period effects than items relating to prices in gas and fossil fuels, both of which were topics of heated public debate at the time of the survey due to the war between Russia and Ukraine.

The central independent variables, social recognition relating to class and generation, are captured by four newly formulated items which are all measured with a 5-point Likert scale and cover different core aspects of recognition. Two items were similar for both categories as people were asked whether they feel that people of their generation or class were taken seriously and whether the opinions of people of their generation or class were respected. The further items were worded more specifically to precisely capture the relevant dimensions of respect relating to either social category. For social class, people were asked whether they feel that people of their class receive enough respect for the work they do, and whether they are treated as an equal part of society. For recognition for generation, people were asked whether they think that the fears and needs of their generation are considered sufficiently.

Hypotheses 4–7 state that the relationship between social recognition and pro-environmental attitudes is constituted by specific mediators.

The first mediator, which is general trust, is captured by a single item measured on an 11-category scale.

The second mediator, trust in political institutions, combines four items which are measured on a 5-category scale. While two of these measures capture trust in the chambers of the current German legislation which are the central government and federal council, the remaining two items measure trust in political parties and politicians in Germany on a 5-category scale.

The third mediator, subjective financial situation, is captured via self-assessed financial situation and financial security, both of which are measured on a 5-category scale. Lastly, solidarity is measured with a variable that combines three self-developed items measured on a 5-category scale which capture willingness to support people who are less wealthy, have an immigration background, or depend on social welfare.

Demographic controls used in all regression analyses are gender (male or female), immigration background, socialization in the eastern or western part of Germany, and parenthood (does the respondent have children–yes or no). Socio-economic controls include current occupational situation (e.g., employed, retired, care work), employment status (e.g., worker, self-employed, employee), and level of education (International Standard Classification of Education; ISCED 2011).

To measure recognition for different social categories, respondents were asked to self-categorize into a social class and generation. These variables were included when testing differences in effects of recognition for different classes and generations (hypotheses 2 and 3). Originally, the social class measure was composed of eight groups including *lower class*, *ordinary people*, *working class*, *self-employed*, *academics*, *entrepreneurs*, *upper class*, and *middle class*. Because some of the subgroups were too small to conduct meaningful analyses and because some of the nuanced distinctions between groups were not necessary for the analyses, *ordinary people* were subsumed with the *lower-class* category and *academics* as well as *entrepreneurs* were assigned to the *upper-class* category. The group of *self-employed* people does not fit with any other of the categories and there is no specific assumption about the influence of social recognition on this group, which is why it is included in the analysis for completeness only.

The generation measure was originally composed of five groups that included *Generation until 1945*, *Post-war Generation*, *Generation of 1968*, *Baby boomers*, and one category for *Generation X/Y and Z*. Although these categories are commonly associated with specific birth cohorts (in the german context the birth cohorts are associated as follows: *Generation until 1945*: Born until 1945; *Generation of 1968*: 1940–1950, *Post-war Generation*: 1946–1955; *Baby boomers*: 1956–1965; *Generations X/Y/Z*: 1966–2009 [60]), we decided not to indicate numbers in the survey. This decision was based on the fact that we wanted to avoid people simply stating their age and instead tried to capture membership in the categories that are meaningful for them. This is also the reason some generation categories are overlapping.

As the *Generation of 1945* group was too small to allow for meaningful analyses, it was subsumed with the *Post-war Generation* into one *War-Generation* category.

All variables measured with multiple items are included as sum-scores in the regression analyses and as factors in the structural equation model (there is so far no consensus on how to conduct explorative factor analysis in multiple imputation, which is why we rely on sum-scores in the regression analyses. Preliminary factor analysis based on the observed data showed that this approach is appropriate). For all variables, the distributions and questions are listed in Appendix A (Tables A1–A3).

### 3.4. Models

Hypothesis 1 is tested by three linear regression models per social category, starting with a regression including only the recognition variable and subsequently adding more of the control variables. Hypotheses 2 and 3 are estimated by including social class and generational belonging and their interactions with recognition to the third and fourth regression. To gain first insights into the importance of the mediators, the following models display the effects of social recognition after each of the mediators is added to the third regression.

For reasons of multicollinearity and in building on the results of the regression analyses the number of variables is reduced in the SEM. For the independent variables, only social class recognition is included because even though the effects of both kinds of recognition are similar, the influence of generational recognition is not as strong. Regarding the control variables, only gender and parenthood are considered. Further, the measures of subjective financial situation are reduced to the self-assessed financial situation.

As a robustness check, the model is then estimated separately for each indicator of support for climate change measures.

## 4. Results

The first research question of this paper is whether social recognition has an influence on climate change mitigation measures. According to hypothesis 1, we expect that the feeling of being recognized has a positive effect on evaluating climate protection measures and that this effect goes above and beyond demographic characteristics as well as economic features. We expect this to be the case for social class recognition as well as recognition for belonging to a generation.

### 4.1. Figure 1: The Influence of Social Recognition on Climate Change Mitigation Measures

Results in Figure 1 are given in standard deviations and show that feeling recognized for being part of a generation as well as social class has a significant and positive influence on support for measures that are aimed at climate protection (see Appendix A, Tables A4 and A5).

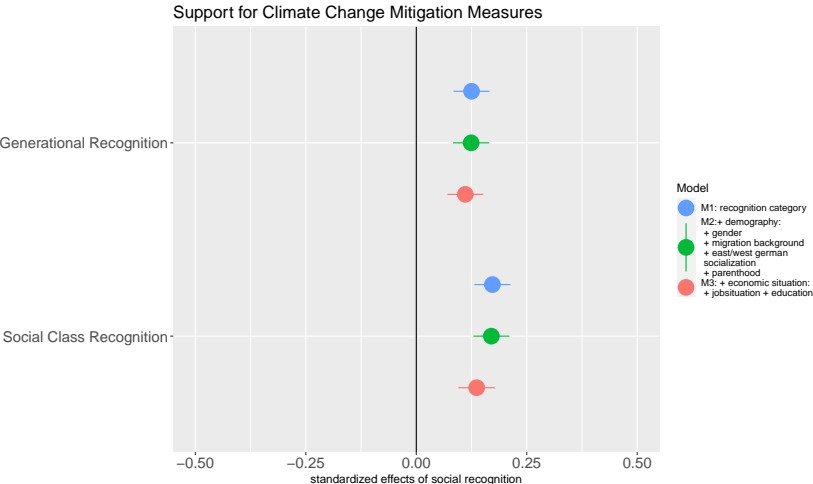

**Figure 1.** Effect of social recognition indicators on climate change mitigation measures.

This effect remains almost unchanged after adding the demographic variables and changes more strongly only in cases of social class recognition after indicators of economic situation are added. In comparing the standardized effect sizes, we see that social class recognition has a stronger influence on support for climate protection measures than recognition for belonging to a generation.

### 4.2. *Figure 2: The Different Role of Recognition for Different Generations and Classes*

Hypothesis 2 stated that recognition should only influence support for climate change mitigation measures for older generations because it strengthens solidarity and that the influence should be considerably smaller for younger generations. This hypothesis is tested by including interaction effects into the model. The results as displayed in Figure 2 show an interaction between belonging to a generation and feeling recognized for it. They indicate that whereas social recognition has a positive influence for older generations, this relationship does not exist for younger generations. The interaction is significant, as can be seen in the coefficient table (see Appendix A, Table A6).

**Figure 2.** Interaction effects of recognition for generation on support for climate change mitigation measures.

Hypothesis 3 stated a difference in the influence of social recognition on support for climate protection measures for the lower and working class on the one hand and middle and upper class on the other. Yet the results do not show any class-specific interaction (see Appendix A, Table A7). Therefore, hypothesis 3, which suggested that social recognition can remedy feelings of perceived disrespect for the lower and working classes, is not supported by our data. Instead, we see that feelings of respect strengthen willingness to support climate change mitigation measures for all classes. According to our argument this implies that the experience of respect compensates for fear of economic loss or restriction of consumption and behavior just the same for the self-employed, middle, and upper classes. This is plausible, as an increase in prices as well as other restrictions would also cause inconvenience to those who do not have to fear for their subsistence. It also shows that within this data there is no indication of the lower and working classes feeling especially alienated from the discourse, which according to hypothesis 3 would result in a stronger influence of respect on their attitudes towards climate policies.

Up to now, the results support the general assumption that social recognition has a positive effect on support for climate protection measures but do not provide further information about potential mechanisms through which social recognition might operate. The following models include all potential mediators and allow us to obtain a better idea about the way that social recognition influences pro-environmental attitudes.

### 4.3. Figure 3: First Insights Regarding Mediators

To avoid multicollinearity and because of the stronger impact of social class recognition, the following analyses only include social class recognition as an independent variable. In displaying the estimated effects of social class recognition in standard deviations, Figure 3 shows the impact of social class recognition in five regression analyses, of which the first includes the recognition indicator and the control variables (see also Appendix A, Table A8).

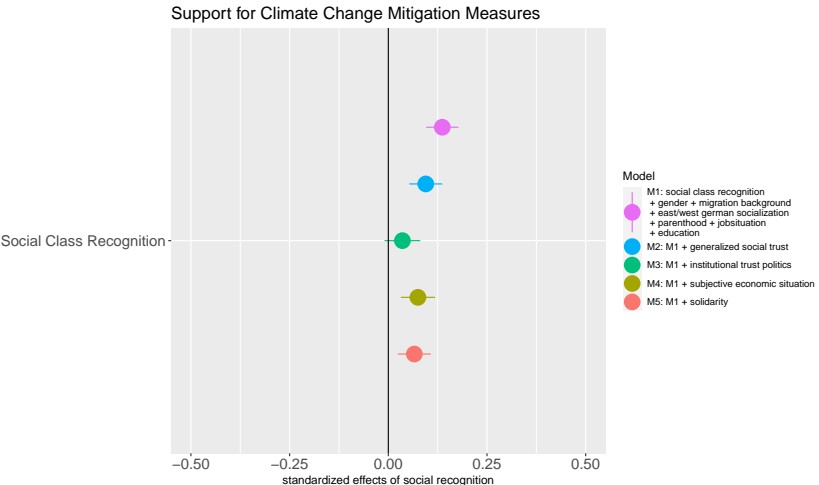

**Figure 3.** Effects of mediators on climate change mitigation measures.

This first model is then complemented with one of the suggested mediators in each model separately. As the coefficient size decreases in all models that include mediators, the graph shows that all indicators lead to a decrease in influence of social class recognition on support for measures. While the influence is strongest in cases of the institutional trust indicator, it is smallest for the indicator measuring generalized trust. While these results do support our hypotheses, testing our assumptions through regression analyses alone has its limitations. Regression analyses do not allow us to test the strength and significance of mediation effects or to make comparisons by testing different pathways at the same time.

They also do not allow us to make statements about relationships between explanatory variables or whether the suggested theoretical model fits the observed data in an appropriate way. For these reasons, we proceed by estimating a structural equation model.

### 4.4. Figure 4: The Direct and Indirect Effects of Social Class Recognition

With an excellent model fit (CFI: 0.98; TLI: 0.97; SRMR: 0.03), the relationships as defined in the model displayed in Figure 4 can be understood to be an adequate representation of the relationships that are found in the data.

In addition to modeling the relationships between central variables, mediators, and control variables, the model allows for correlations between the error terms of the mediating variables. This decision accounts for the fact that assuming all relationships between mediators are fully covered by the effect of social recognition on each of them is questionable.

Political institutions play a central role in allocating resources and shaping relationships between people, which makes it likely that there is some correlation between trust in political institutions, feelings of financial stability, as well as trusting others and willingness to help which is not explained by social recognition. In addition, it is reasonable to assume that trust in others as well as assessment of financial situation are related to willingness to help people in a less fortunate position. One further specification concerns the factor that measures trust in political institutions. This factor includes two items which relate to the current governing institutions and two that relate to political parties and actors in general. As political parties and actors in Germany differ greatly in their attitude towards climate protection, it is reasonable to assume that there is a difference between trust in governing bodies and actors and general trust in politics.

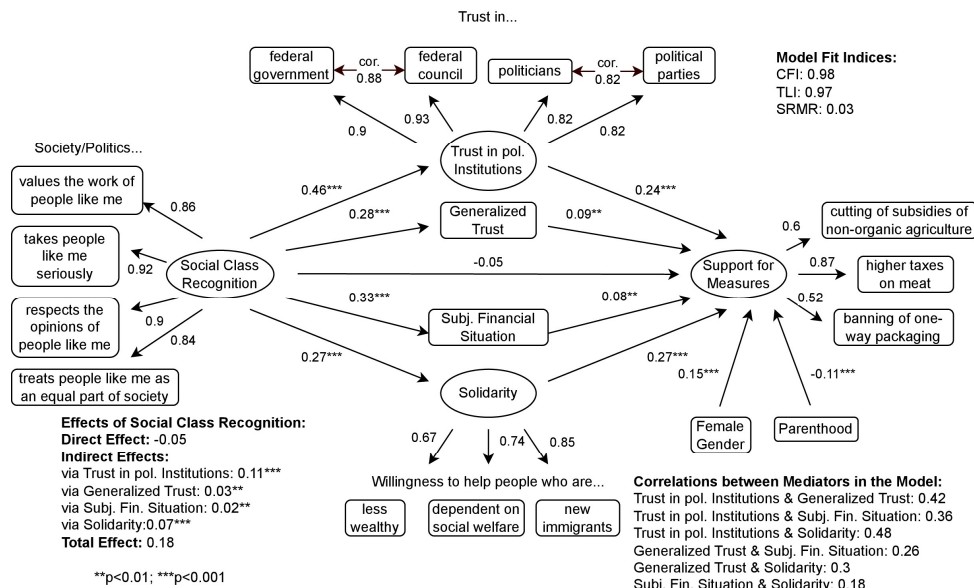

**Figure 4.** Structural Equation Model displaying the direct and indirect effects of social class recognition through the mediators.

Starting with the control variables, we see that women are more in favor of measures and people with children are less in favor of measures. Whereas the former adds information to an on-going debate on whether gender is an important predictor for support of climate protection (e.g., [8,9]), the latter finding is surprising when assuming that people care more about the future when it affects their next of kin. It probably has to be understood with respect to the costs of childcare and the increase in costs that most measures imply.

The direct effect of social recognition on support for climate change measures is negative, yet very small and not statistically significant, which implies that it is completely mediated. All the coefficients reflecting the correlation between social recognition and the potentially mediating variables are statistically significant at a one percent level and go in the expected direction. In alignment with our hypotheses, an increase in social recognition leads to a significant increase in trust in political institutions, strengthens generalized trust and solidarity, and leads to a positive assessment of the financial situation. This influence is strongest in cases of trust in political institutions and smallest for solidarity, with a difference of about 0.2 standard deviations.

The second part of the mediation shows that the direction of coefficients is also in line with hypotheses and statistically significant; this implies that there is an influence of social recognition on support for climate protection measures which operates through the suggested mediators. All in all, the total effect of social recognition equates to an effect of 0.18 standard deviations, which lends further support to hypothesis 1 that stated the general importance of social recognition for support for climate protection measures. In putting attention on the individual pathways through which social recognition affects support for climate protection measures indirectly, we see that the effect is largest for trust in political institutions and solidarity.

### 4.5. Robustness Check

To back the findings of the previous section, the SEM is calculated separately for each of the three single item measures as the dependent variables. While this section only presents the central findings, all graphs and coefficients can be found in Appendix A (see Figures A1–A3).

Regarding the control variables, we find that women are more in favor with two of three policies and that parenthood decreases support when it comes to a higher taxation on meat, whereas it does not affect cutting of subsidies for non-organic agriculture or

prohibiting one-way packaging. Whereas the mixed results for gender are in line with previous research, the second result is interesting, as it rebuts the previous assumption that people reject measures mostly for financial reasons because they are not against an increase in food prices due to less subsidies for non-organic food.

The indirect effects of social recognition are all significant in the factor model and in the two single-item models that refer to price increases through either higher taxation on meat or cutting of subsidies for non-organic agriculture, which corroborates hypotheses 4–7. Regarding the fourth model which measures support for prohibiting one-way packaging, however, hypotheses 3 and 5 must be rejected as neither subjective financial situation nor generalized trust are statistically significant influence factors.

The indirect effect of social recognition is consistently strongest and significant when it is mediated via trust in political institutions and solidarity.

## 5. Summary of Results

In our analyses, we find that the respect that people receive for being part of a social class and generation strengthens support for climate protection and that this effect goes beyond demographic and economic characteristics. We further find that the effect of social class recognition being stronger than that of generational belonging is likely because respect does not have an influence on support for climate protection for younger people. This corroborates our assumption that while younger generations support climate protection out of self-interest, respect is an important influence factor for older generations because it strengthens solidarity.

There is no support for the assumption that the effect of social recognition should vary between social classes, which objects to the idea that lower- and working-class people in particular are less willing to contribute to climate protection because of feeling disrespected. Instead, feeling respected has a positive influence across all social classes.

In looking closer at the way that social recognition operates, we find partial evidence that it influences attitudes through the four mechanisms that both our and previous research suggest. Assessment of financial situation, trust in others and in political institutions, and solidarity are all positively influenced by social recognition. We find that this influence is consistently significant when it operates through institutional trust and solidarity but is less strong and in some cases not significant for subjective financial situations and generalized trust.

The next and final section of this paper is dedicated to putting these results into a political and social context and to addressing the limitations of our research. It closes with some recommendations for future research.

## 6. Discussion

Scientists and most of the public agree: the issue of global warming is worrying and calls for urgent action [61]. Politicians worldwide are making considerable efforts to reach agreements and even though many measures to limit global warming have been successfully installed over the last decades [1], leading climate research institutions state that these measures are insufficient [1,59] and that currently none of the social drivers of climate protection have been developed to their full potential to limit global warming [59]. Necessary progress is slowed down to a considerable degree because support for actual measures is disputed [8,32,62], which calls for an increased inquiry into what makes people support said measures. Our paper contributes to the current debate by introducing social recognition as an influential factor. Drawing on social identity theory [42], we find that social recognition for social class and generation are important influence factors on support for climate change mitigation measures and that social class recognition has an influence on other factors that have been established as relevant for the support for climate protection by previous research [4,31,55]. Assessment of financial situation, trust in others and in institutions, and solidarity all relate to the costs and collective good character of climate protection and are all positively influenced by social recognition. The fact that the influence of social recognition is consistently significant and strong when it operates through institutional trust and solidarity as opposed to being weaker and,

in some cases, insignificant when it operates through subjective financial situations and generalized trust invites us to think about the societal dimensions that all variables are embedded in. Solidarity and trust in political institutions as well as support for climate change measures are concepts that point to the macro-level, whereas generalized trust and individual financial situation refer to relationships on the individual level. Social recognition as respect for one's group is located in between these layers on the meso-level, which explains why we find that it is a strong influence factor for all mediating variables. Yet the second path of the relationship connecting the mediators and the dependent variable varies clearly in strength between the topics that relate to either level. We suggest that this is because the topic of climate protection is unambiguously one that needs to be addressed in a common effort, which makes concepts relating to institutions and large-scale groups much more important than aspects relating to individual situations or other individuals, even if they are referred to as general others. Against this background it can be assumed that support for measures can likely be strengthened when it is framed and addressed as a common project that needs the contribution of everybody and is coordinated in a way that does not hurt expectations of solidarity with those who might have less to contribute. This thought is well in line with research indicating that measures are at times disputed when people fear unjust treatment [8] and that measures are more likely to be supported when they are perceived to be fair [4,63–65]. Building on this, it would be promising for further research to investigate the relationships between social recognition, solidarity, and fairness expectations and relate all of them to the support for climate protection. While we do think that the results we present shed light on an important and, so far, understudied aspect in the debate on climate protection, we must also acknowledge some limitations of our study.

First, and relating to the generation categories, we think that it would be promising to conduct a study that differentiates between the youngest three generations, which are generation X, Y, and Z. While we do find significant differences between those belonging to the generation X/Y/Z category and those belonging to categories of older generations, it would be interesting to look more closely into whether the effect of recognition on the youngest generation, which will on average be most affected by global warming, is different from the effect on others.

Second, and relating to the specialty of the German case, it would be important to take a closer look at differences in attitudes between people who grew up in either the eastern or western parts of Germany. While we do control for differences between socialization in East and West Germany and find that people who grew up in West Germany are more supportive of our climate measures, it is beyond the scope of this paper to address this in greater detail. It would, however, be promising to do so at another time, as perceived disrespect is also a recurring theme in the discussion of attitudinal differences between the previously divided parts of Germany.

Third, and regarding the dependent variable, it could be argued that the focus on food consumption is too narrow. There are other important and highly salient topics in the discourse such as costs of transportation and carbon taxation (e.g., [6]) that we do not cover, and which would be important to address in future research. Against the background of the current armed conflicts and the associated impacts on the environment that are caused by the expansion of the military, the consequences of war, and the reconstruction of destroyed areas, it is also highly recommended for future research to analyze whether and how people assess military expenditure and participation in said conflicts with regards to climate protection.

Fourth, for future research it would also be promising to look more closely into the question of what really does make people feel respected as being part of their group and therefore in supporting climate protection. While data like ours that are derived from a cross-sectional survey cannot account for causal effects, this could be an interesting topic for experimental studies.

Finally, given the strong influence of solidarity on support for climate change mitigation measures, it would be important to learn more about the impact of additional influence factors on solidarity.

All in all, our results show that attitudes towards climate protection measures are clearly influenced by factors that are also central to research on societal cohesion. The

fact that social recognition has a significant effect on each of these influence factors is an invitation for future research to think about the role that respect for social group belonging may play in the explanation of further societal phenomena and collective good problems.

**Author Contributions:** Methodology, software, data curation, visualization, writing—review and editing: S.J. and M.G.; conceptualization, formal analysis, investigation, writing—original draft preparation: S.J.; resources, supervision, project administration: M.G. All authors have read and agreed to the published version of the manuscript.

**Funding:** This research was funded by the Federal Ministry of Education and Research (BMBF) of Germany under the Grant Number 01UG2107. We acknowledge support from the Open Access Publication Fund of the University of Tübingen.

**Institutional Review Board Statement:** All subjects gave their informed consent for inclusion before they participated in the study. The study was conducted in accordance with the Declaration of Helsinki. The study was approved by the Faculty of Economics and Social Sciences Ethics Committee (File Reference AZ. A2.5.4-232_aa; accepted on 4 July 2022).

**Informed Consent Statement:** Informed consent was obtained from all subjects involved in the study.

**Data Availability Statement:** The data presented in this study are available upon request from the corresponding author. The data are not publicly available at this stage of our project due to participant confidentiality, as the multiple imputation approach requires full data access for replication of our results. If the editors of Sustainability deem this article appropriate for the journal, we will make the dataset available following a strict anonymization strategy.

**Conflicts of Interest:** The authors declare no conflict of interest.

## Appendix A

*Appendix A.1. Variable Wordings and Distributions*

**Table A1.** Mean, Standard Deviation, and Factor Loadings Based on Observed Data.

| Variable | Wording | Mean | sd | Factor Loadings | n |
|---|---|---|---|---|---|
| Climate change mitigation measures | Financial support for non-organic farming should be cut even if that means noticeable increases of food prices. (In SEM: Cutting of subsidies for non-organic agriculture) | 2.38 | 1.17 | 0.61 | 2048 |
| | Taxation of meat products should increase significantly. (In SEM: Higher taxes on meat) | 2.55 | 1.31 | 0.85 | 2047 |
| | One-way packaging should not be allowed even if it means that taking away food and beverages gets much more complicated. (In SEM: Banning of one-way packaging) | 3.55 | 1.18 | 0.52 | 2047 |
| Social Recognition | | | | | |
| Class | Society/Politics… | | | | |
| | …values the work of people like me. | 2.68 | 1.09 | 0.86 | 3046 |
| | …takes people like me seriously. | 2.68 | 1.1 | 0.92 | 3046 |
| | …respects the opinions of people like me. | 2.73 | 1.1 | 0.90 | 3046 |
| | …treats people like me as an equal part of society. | 2.95 | 1.16 | 0.84 | 3046 |
| Generation | Society/Politics… | | | | |
| | …takes people of my generation seriously. | 2.87 | 1.01 | 0.86 | 2050 |
| | …takes the fears of people of my generation into account. | 2.6 | 1 | 0.89 | 2049 |
| | …respects the opinions of people of my generation. | 2.85 | 1.01 | 0.87 | 2049 |
| | …respects the needs of people of my generation. | 2.62 | 0.97 | 0.83 | 2047 |
| Solidarity | | | | | |

**Table A1.** *Cont.*

| Variable | Wording | Mean | sd | Factor Loadings | n |
|---|---|---|---|---|---|
| Willingness to help others | Would you be willing to support another person that… | | | | |
| | …is less wealthy than you? | 3.6 | 0.89 | 0.67 | 1985 |
| | …is dependent of social welfare? | 2.89 | 1.06 | 0.77 | 1981 |
| | …has newly immigrated to Germany? | 2.82 | 0.91 | 0.78 | 1984 |
| | Trust | | | | |
| Trust in Political Institutions | How much do you trust… | | | | |
| | …the federal government? | 2.32 | 1.08 | 0.90 | 1987 |
| | …the federal council? | 2.39 | 1.08 | 0.91 | 1979 |
| | …politicians? | 2.20 | 0.92 | 0.87 | 1965 |
| | …political parties? | 2.05 | 0.93 | 0.86 | 1985 |
| Generalized Trust | Some people say that most people are to be trusted. Others think that one cannot be cautious enough. One cannot be cautious enough (1)—Most people can be trusted (11) | 4.96 | 2.57 | | 3046 |
| | Assessment of Financial Situation | | | | |
| Financial Security | How financially secure do you feel in life? Not at all secure (1)—Completely secure (11) | 6.07 | 2.54 | | 3046 |
| Financial Situation | How would you assess your financial situation today? Very bad (1)—very good (5) | 3.05 | 0.94 | | 3046 |

**Table A2.** Variable Wording and Distribution of Categorical Variables Based on Observed Data.

| Variable | Category | n | N |
|---|---|---|---|
| Demography | | | |
| Gender | | | 3046 |
| | male | 1548 | |
| | female | 1498 | |
| | other | 0 | |
| Immigration Background | | | 3040 |
| | No immigration background | 2644 | |
| | Parents are migrants | 239 | |
| | Respondent is migrant | 6 | |
| | Parents and respondent are migrants | 151 | |
| | No reply | 6 | |
| Grew up in eastern or western Germany | | | 3023 |
| | East | 1068 | |
| | West | 1783 | |
| | Neither | 124 | |
| | Other | 48 | |
| | No reply | 23 | |

**Table A2.** *Cont.*

| Variable | Category | n | N |
|---|---|---|---|
| Socio-economic controls | | | |
| ISCED 2011 | | | 2968 |
| | ISCED_24 | 121 | |
| | ISCED_34 | 99 | |
| | ISCED_35 | 1499 | |
| | ISCED_45 | 44 | |
| | ISCED_65 | 363 | |
| | ISCED_64 | 353 | |
| | ISCED_74 | 437 | |
| | ISCED_84 | 52 | |
| | No information | 78 | |
| Job-Situation | | | 2946 |
| | Worker | 198 | |
| | Academic | 50 | |
| | Self-employed | 115 | |
| | Civil Servant | 150 | |
| | Executing Staff | 239 | |
| | Qualified Staff | 520 | |
| | Independent Staff | 432 | |
| | Managing Staff | 40 | |
| | Staff (no details) | 47 | |
| | Other | 26 | |
| | School | 104 | |
| | Unemployed | 48 | |
| | Unable | 87 | |
| | Retired | 796 | |
| | Carework | 94 | |
| | No reply to job status | 92 | |
| | No reply to occupation | 8 | |

**Table A3.** Variable Wording and Distribution of Social Categories Based on Observed Data.

| Variable | Category | n | N |
|---|---|---|---|
| Social Categories | | | |
| Social Class (original) | | | 2965 |
| | Lower Class | 92 | |
| | Ordinary People | 394 | |
| | Working Class | 684 | |
| | Self-employed | 152 | |
| | Academic | 392 | |
| | Entrepreneur | 14 | |

**Table A3.** *Cont.*

| Variable | Category | n | N |
|---|---|---|---|
| | Upper Class | 24 | |
| | Middle Class | 1213 | |
| | Other | 47 | |
| | No reply | 34 | |
| Social Class (combined as in analyses) | | | |
| | Lower Class | 486 | 2965 |
| | Working Class | 684 | |
| | Self-employed | 152 | |
| | Upper Class | 430 | |
| | Middle Class | 1213 | |
| | Other | 47 | |
| | No reply | 34 | |
| Generation (original) | | | 1682 |
| | Generation until 45 | 23 | |
| | Post-war Generation | 310 | |
| | Generation of 68 | 288 | |
| | Baby Boomer | 385 | |
| | Generation X/Y/Z | 619 | |
| | Other | 19 | |
| | No reply | 350 | |
| | No information because of planned missing design | 995 | |
| Generation (combined as in analyses) | | | 1682 |
| | War-generation | 333 | |
| | Generation of 68 | 288 | |
| | Baby Boomer | 385 | |
| | Other | 19 | |
| | No reply | 350 | |
| | No information because of planned missing design | 995 | |

*Appendix A.2. Regression Tables*

Appendix A.2.1. Regression Model Including Recognition for Generation

**Table A4.** Standardized coefficients of independent variables and significant control variables, standard errors, and *t*-values. The model additionally includes control variables for gender, East/West German provenance, migration status, job situation, and ISCED 2011.

| | Support for Measures | | |
|---|---|---|---|
| | Model 1 | Model 2 | Model 3 |
| (Intercept) | 0.00001 | −0.184 *** | −0.533 *** |
| | 0.022 | 0.043 | 0.136 |
| | 0.0005 | −4.286 | −3.903 |

**Table A4.** *Cont.*

| | Support for Measures | | |
| --- | --- | --- | --- |
| | **Model 1** | **Model 2** | **Model 3** |
| Recognition for Generation | 0.125 *** | 0.124 *** | 0.111 *** |
| | 0.021 | 0.021 | 0.021 |
| | 5.999 | 5.937 | 5.358 |
| Female | | 0.112 ** | 0.130 ** |
| | | 0.042 | 0.043 |
| | | 2.647 | 3.014 |
| West German | | 0.127 ** | 0.139 ** |
| | | 0.043 | 0.043 |
| | | 2.944 | 3.201 |
| No children | | 0.169 *** | 0.155 *** |
| | | 0.040 | 0.042 |
| | | 4.202 | 3.666 |
| ISCED_34 | | | 0.422 ** |
| | | | 0.161 |
| | | | 2.615 |
| ISCED_64 | | | 0.260 * |
| | | | 0.126 |
| | | | 2.067 |
| ISCED_74 | | | 0.364 ** |
| | | | 0.127 |
| | | | 2.859 |
| ISCED_84 | | | 0.416 * |
| | | | 0.195 |
| | | | 2.137 |
| Qualified Staff | | | 0.293 ** |
| | | | 0.097 |
| | | | 3.007 |
| Independent Staff | | | 0.292 ** |
| | | | 0.101 |
| | | | 2.898 |
| Retired | | | 0.292 ** |
| | | | 0.091 |
| | | | 3.195 |
| Carework | | | 0.288 * |
| | | | 0.141 |
| | | | 2.037 |
| Num.Obs. | 3046 | 3046 | 3046 |
| Num.Imp. | 30 | 30 | 30 |
| R2 | 0.016 | 0.032 | 0.073 |
| R2 Adj. | 0.015 | 0.029 | 0.062 |

* $p < 0.05$, ** $p < 0.01$, *** $p < 0.001$.

Appendix A.2.2. Regression Model Including Recognition for Social Class

**Table A5.** Standardized coefficients of independent variables and significant control variables, standard errors, and *t*-values. The model additionally includes control variables for gender, East/West German provenance, migration status, job situation, and ISCED 2011.

| | Support for Measures | | |
| --- | --- | --- | --- |
| | **Model 1** | **Model 2** | **Model 3** |
| (Intercept) | $-2 \times 10^{-17}$ | $-0.174$ *** | $-0.481$ *** |
| | 0.022 | 0.043 | 0.136 |
| | $-7 \times 10^{-16}$ | $-4.021$ | $-3.537$ |
| Social Class Recognition | 0.172 *** | 0.170 *** | 0.137 *** |
| | 0.021 | 0.021 | 0.021 |

**Table A5.** *Cont.*

|  | Support for Measures | | |
|---|---|---|---|
|  | **Model 1** | **Model 2** | **Model 3** |
|  | 8.306 | 8.189 | 6.514 |
| Female |  | 0.113 ** | 0.125 ** |
|  |  | 0.042 | 0.043 |
|  |  | 2.686 | 2.914 |
| Parents and Self Migrants |  | −0.209 | −0.238 * |
|  |  | 0.119 | 0.117 |
|  |  | −1.752 | −2.024 |
| West German |  | 0.120 ** | 0.132 ** |
|  |  | 0.043 | 0.044 |
|  |  | 2.757 | 3.042 |
| No children |  | 0.159 *** | 0.149 *** |
|  |  | 0.040 | 0.042 |
|  |  | 3.959 | 3.517 |
| ISCED_34 |  |  | 0.373 * |
|  |  |  | 0.161 |
|  |  |  | 2.323 |
| ISCED_74 |  |  | 0.309 * |
|  |  |  | 0.128 |
|  |  |  | 2.420 |
| Qualified Staff |  |  | 0.281 ** |
|  |  |  | 0.097 |
|  |  |  | 2.913 |
| Independent Staff |  |  | 0.273 ** |
|  |  |  | 0.100 |
|  |  |  | 2.732 |
| Retired |  |  | 0.269 ** |
|  |  |  | 0.090 |
|  |  |  | 2.973 |
| Carework |  |  | 0.297 * |
|  |  |  | 0.141 |
|  |  |  | 2.113 |
| Num.Obs. | 3046 | 3046 | 3046 |
| Num.Imp. | 30 | 30 | 30 |
| R2 | 0.030 | 0.046 | 0.079 |
| R2 Adj. | 0.029 | 0.042 | 0.068 |

* $p < 0.05$, ** $p < 0.01$, *** $p < 0.001$.

Appendix A.2.3. Regression Model Including Recognition for Generation and Interaction Effects

**Table A6.** Standardized coefficients of independent variables and significant control variables, standard errors, and *t*-values. The model additionally includes control variables for gender, East/West German provenance, migration status, job situation, and ISCED 2011.

|  | Support for Measures | | | | |
|---|---|---|---|---|---|
|  | **Model 1** | **Model 2** | **Model 3** | **Model 4** | **Model 5** |
| (Intercept) | 0.080 * | 0.096 * | 0.077 * | −0.121* | −0.481 *** |
|  | 0.038 | 0.038 | 0.038 | 0.057 | 0.139 |
|  | 2.096 | 2.521 | 2.022 | −2.109 | −3.460 |
| Pre and Post Wargeneration | −0.115 | −0.143 * | −0.132 * | −0.078 | −0.061 |
|  | 0.067 | 0.066 | 0.066 | 0.069 | 0.078 |
|  | −1.723 | −2.153 | −1.994 | −1.135 | −0.782 |
| Generation of 68 | −0.036 | −0.074 | −0.066 | −0.025 | 0.026 |
|  | 0.070 | 0.070 | 0.070 | 0.072 | 0.075 |
|  | −0.520 | −1.054 | −0.937 | −0.354 | 0.351 |
| Babyboomer | 0.0002 | −0.029 | −0.014 | 0.019 | 0.044 |
|  | 0.064 | 0.063 | 0.063 | 0.065 | 0.065 |

**Table A6.** *Cont.*

| | Support for Measures | | | | |
|---|---|---|---|---|---|
| | **Model 1** | **Model 2** | **Model 3** | **Model 4** | **Model 5** |
| Other Generation | 0.003 | −0.453 | −0.220 | 0.295 | 0.681 |
| | −0.135 | −0.179 | −0.238 | −0.185 | −0.220 |
| | 0.232 | 0.230 | 0.237 | 0.237 | 0.234 |
| | −0.581 | −0.778 | −1.006 | −0.779 | −0.941 |
| No Reply Generation | −0.316 *** | −0.331 *** | −0.312 *** | −0.280 *** | −0.196 ** |
| | 0.066 | 0.065 | 0.065 | 0.067 | 0.067 |
| | −4.808 | −5.075 | −4.787 | −4.198 | −2.920 |
| Recognition for Generation | | 0.127 *** | −0.025 | −0.019 | −0.016 |
| | | 0.021 | 0.037 | 0.038 | 0.037 |
| | | 6.068 | −0.662 | −0.500 | −0.424 |
| Pre and Post Wargeneration × Recognition | | | 0.239 *** | 0.233 *** | 0.213 ** |
| | | | 0.067 | 0.066 | 0.066 |
| | | | 3.589 | 3.523 | 3.246 |
| Generation of 68 × Recognition | | | 0.216 ** | 0.201 ** | 0.180 ** |
| | | | 0.069 | 0.069 | 0.068 |
| | | | 3.153 | 2.931 | 2.651 |
| Babyboomer × Recognition | | | 0.191 ** | 0.173 ** | 0.146 * |
| | | | 0.063 | 0.063 | 0.063 |
| | | | 3.016 | 2.730 | 2.313 |
| No Reply Generation × Recognition | | | 0.205 ** | 0.191 ** | 0.187 ** |
| | | | 0.067 | 0.067 | 0.066 |
| | | | 3.050 | 2.846 | 2.815 |
| Planned Missing × Recognition | | | 0.174 ** | 0.167 ** | 0.142 * |
| | | | 0.059 | 0.059 | 0.059 |
| | | | 2.964 | 2.851 | 2.414 |
| Female | | | | 0.120 ** | 0.134 ** |
| | | | | 0.043 | 0.043 |
| | | | | 2.800 | 3.088 |
| West German | | | | 0.117 ** | 0.127 ** |
| | | | | 0.043 | 0.043 |
| | | | | 2.702 | 2.941 |
| No Children | | | | 0.145 *** | 0.143 ** |
| | | | | 0.042 | 0.043 |
| | | | | 3.442 | 3.296 |
| ISCED_34 | | | | | 0.386 * |
| | | | | | 0.162 |
| | | | | | 2.381 |
| ISCED_74 | | | | | 0.335 ** |
| | | | | | 0.128 |
| | | | | | 2.611 |
| Qualified Staff | | | | | 0.293 ** |
| | | | | | 0.098 |
| | | | | | 2.997 |
| Independent Staff | | | | | 0.290 ** |
| | | | | | 0.101 |
| | | | | | 2.881 |
| Retired | | | | | 0.285 ** |
| | | | | | 0.094 |
| | | | | | 3.033 |
| Num.Obs. | 3046 | 3046 | 3046 | 3046 | 3046 |
| Num.Imp. | 30 | 30 | 30 | 30 | 30 |
| R2 | 0.009 | 0.025 | 0.033 | 0.048 | 0.084 |
| R2 Adj. | 0.007 | 0.023 | 0.029 | 0.040 | 0.069 |

* $p < 0.05$, ** $p < 0.01$, *** $p < 0.001$.

## Appendix A.2.4. Regression Model Including Social Class Recognition and Interaction Effects

**Table A7.** Standardized coefficients of independent variables, interaction effects and significant control variables, standard errors, and *t*-values. The model additionally includes control variables for gender, East/West German provenance, migration status, job situation, and ISCED 2011.

| | Support for Measures | | | | |
| --- | --- | --- | --- | --- | --- |
| | **Model 1** | **Model 2** | **Model 3** | **Model 4** | **Model 5** |
| (Intercept) | −0.121 ** | −0.094 * | −0.102* | −0.263 *** | −0.459 ** |
| | 0.046 | 0.046 | 0.046 | 0.060 | 0.139 |
| | −2.645 | −2.052 | −2.195 | −4.366 | −3.303 |
| Lower Class | −0.101 | −0.053 | −0.062 | −0.081 | −0.115 |
| | 0.066 | 0.066 | 0.072 | 0.072 | 0.075 |
| | −1.533 | −0.795 | −0.864 | −1.124 | −1.540 |
| Self-employed | 0.169 | 0.138 | 0.144 | 0.153 | 0.076 |
| | 0.102 | 0.102 | 0.102 | 0.101 | 0.116 |
| | 1.656 | 1.358 | 1.411 | 1.505 | 0.655 |
| Upper Class | 0.402 *** | 0.301 *** | 0.316 *** | 0.306 *** | 0.032 |
| | 0.068 | 0.069 | 0.075 | 0.074 | 0.087 |
| | 5.948 | 4.386 | 4.238 | 4.124 | 0.367 |
| Middle Class | 0.184 *** | 0.136 * | 0.139* | 0.124 * | 0.031 |
| | 0.055 | 0.055 | 0.055 | 0.055 | 0.058 |
| | 3.338 | 2.455 | 2.510 | 2.252 | 0.537 |
| Other Class | −0.095 | −0.091 | −0.091 | −0.088 | −0.187 |
| | 0.170 | 0.169 | 0.173 | 0.174 | 0.175 |
| | −0.558 | −0.538 | −0.523 | −0.504 | −1.068 |
| No Reply Class | 0.007 | −0.004 | 0.013 | 0.010 | −0.016 |
| | 0.192 | 0.191 | 0.193 | 0.197 | 0.199 |
| | 0.038 | −0.023 | 0.065 | 0.053 | −0.080 |
| Social Class Recognition | | 0.136 *** | 0.096 * | 0.089 * | 0.088 * |
| | | 0.022 | 0.043 | 0.042 | 0.042 |
| | | 6.160 | 2.258 | 2.092 | 2.088 |
| Lower Class × Recognition | | | 0.008 | 0.018 | 0.017 |
| | | | 0.068 | 0.068 | 0.067 |
| | | | 0.114 | 0.271 | 0.249 |
| Self-employed × Recognition | | | 0.119 | 0.133 | 0.111 |
| | | | 0.098 | 0.098 | 0.101 |
| | | | 1.212 | 1.359 | 1.103 |
| Upper Class × Recognition | | | 0.026 | 0.031 | 0.037 |
| | | | 0.070 | 0.069 | 0.069 |
| | | | 0.369 | 0.449 | 0.534 |
| Middle Class × Recognition | | | 0.071 | 0.075 | 0.055 |
| | | | 0.057 | 0.057 | 0.057 |
| | | | 1.236 | 1.308 | 0.972 |
| Other Class × Recognition | | | 0.006 | 0.038 | 0.026 |
| | | | 0.181 | 0.180 | 0.180 |
| | | | 0.035 | 0.208 | 0.147 |
| Class no Reply × Recognition | | | 0.116 | 0.181 | 0.212 |
| | | | 0.239 | 0.238 | 0.239 |
| | | | 0.487 | 0.759 | 0.889 |
| Female | | | | 0.125 ** | 0.129 ** |
| | | | | 0.042 | 0.043 |
| | | | | 2.983 | 3.003 |
| Parents and Self are Migrants | | | | −0.224 | −0.241 * |
| | | | | 0.118 | 0.117 |
| | | | | −1.892 | −2.055 |
| West German | | | | 0.108 * | 0.124 ** |
| | | | | 0.043 | 0.043 |
| | | | | 2.515 | 2.867 |
| No Children | | | | 0.156 *** | 0.154 *** |
| | | | | 0.040 | 0.043 |
| | | | | 3.861 | 3.605 |
| ISCED_34 | | | | | 0.337 * |
| | | | | | 0.163 |
| | | | | | 2.066 |
| ISCED_74 | | | | | 0.264 * |
| | | | | | 0.133 |
| | | | | | 1.979 |

**Table A7.** *Cont.*

| | Support for Measures | | | | |
|---|---|---|---|---|---|
| | **Model 1** | **Model 2** | **Model 3** | **Model 4** | **Model 5** |
| Qualified Staff | | | | | 0.275 ** |
| | | | | | 0.099 |
| | | | | | 2.778 |
| Independent Staff | | | | | 0.268 ** |
| | | | | | 0.103 |
| | | | | | 2.605 |
| Retired | | | | | 0.285 ** |
| | | | | | 0.093 |
| | | | | | 3.069 |
| Carework | | | | | 0.307 * |
| | | | | | 0.143 |
| | | | | | 2.149 |
| Num.Obs. | 3046 | 3046 | 3046 | 3046 | 3046 |
| Num.Imp. | 30 | 30 | 30 | 30 | 30 |
| R2 | 0.025 | 0.042 | 0.044 | 0.060 | 0.083 |
| R2 Adj. | 0.023 | 0.040 | 0.040 | 0.052 | 0.069 |

* $p < 0.05$, ** $p < 0.01$, *** $p < 0.001$.

Appendix A.2.5. Regression Model Including Social Class Recognition and Mediators

**Table A8.** Standardized coefficients of independent variables and significant mediators and control variables, standard errors, and *t*-values. The model additionally includes control variables for gender, East/West German provenance, migration status, job situation, and ISCED 2011.

| | Support for Measures | | | | |
|---|---|---|---|---|---|
| | **Model 1** | **Model 2** | **Model 3** | **Model 4** | **Model 5** |
| (Intercept) | −0.481 *** | −0.418 ** | −0.338 * | −0.372 ** | −0.345 * |
| | 0.136 | 0.135 | 0.134 | 0.135 | 0.135 |
| | −3.537 | −3.101 | −2.518 | −2.756 | −2.549 |
| Social Class Recognition | 0.137 *** | 0.095 *** | 0.036 | 0.075 *** | 0.066 ** |
| | 0.021 | 0.021 | 0.023 | 0.022 | 0.021 |
| | 6.514 | 4.453 | 1.548 | 3.396 | 3.092 |
| Female | 0.125 ** | 0.160 *** | 0.172 *** | 0.168 *** | 0.132 ** |
| | 0.043 | 0.043 | 0.042 | 0.043 | 0.043 |
| | 2.914 | 3.746 | 4.058 | 3.929 | 3.074 |
| Parents and Self Migrants | −0.238 * | −0.234 * | −0.247 * | −0.201 | −0.266 * |
| | 0.117 | 0.116 | 0.115 | 0.116 | 0.115 |
| | −2.024 | −2.016 | −2.151 | −1.731 | −2.303 |
| West German | 0.132 ** | 0.125 ** | 0.099 * | 0.114 ** | 0.088 * |
| | 0.044 | 0.043 | 0.044 | 0.043 | 0.042 |
| | 3.042 | 2.907 | 2.256 | 2.647 | 2.113 |
| No children | 0.149 *** | 0.150 *** | 0.129 ** | 0.151 *** | 0.130 ** |
| | 0.042 | 0.042 | 0.042 | 0.042 | 0.042 |
| | 3.517 | 3.577 | 3.052 | 3.601 | 3.106 |
| ISCED_74 | 0.309* | 0.240 | 0.199 | 0.183 | 0.207 |
| | 0.128 | 0.127 | 0.126 | 0.128 | 0.126 |
| | 2.420 | 1.891 | 1.579 | 1.427 | 1.653 |
| Qualified Staff | 0.281 ** | 0.229 * | 0.184 | 0.208 * | 0.194 * |
| | 0.097 | 0.096 | 0.095 | 0.095 | 0.094 |
| | 2.913 | 2.388 | 1.946 | 2.184 | 2.069 |
| Independent Staff | 0.273 ** | 0.212 * | 0.168 | 0.182 | 0.182 |
| | 0.100 | 0.099 | 0.098 | 0.099 | 0.098 |
| | 2.732 | 2.143 | 1.727 | 1.845 | 1.852 |
| Unable to work | 0.232 | 0.240 | 0.206 | 0.301 * | 0.172 |
| | 0.144 | 0.142 | 0.140 | 0.143 | 0.141 |
| | 1.608 | 1.685 | 1.469 | 2.108 | 1.221 |
| Retired | 0.269 ** | 0.213 * | 0.148 | 0.205 * | 0.192 * |
| | 0.090 | 0.090 | 0.089 | 0.089 | 0.088 |
| | 2.973 | 2.374 | 1.666 | 2.290 | 2.173 |
| Carework | 0.297 * | 0.257 | 0.217 | 0.259 | 0.188 |
| | 0.141 | 0.139 | 0.137 | 0.139 | 0.141 |
| | 2.113 | 1.852 | 1.578 | 1.863 | 1.339 |
| Generalized Trust | | 0.181 *** | 0.126 *** | 0.170 *** | 0.134 *** |

**Table A8.** *Cont.*

| | Support for Measures | | | | |
|---|---|---|---|---|---|
| | Model 1 | Model 2 | Model 3 | Model 4 | Model 5 |
| | | 0.023 | 0.023 | 0.023 | 0.024 |
| | | 7.844 | 5.431 | 7.387 | 5.659 |
| Trust in political Institutions | | | 0.201 *** | | |
| | | | 0.027 | | |
| | | | 7.405 | | |
| Self-assessed financial Situation | | | | 0.083 ** | |
| | | | | 0.028 | |
| | | | | 2.913 | |
| Solidarity | | | | | 0.226 *** |
| | | | | | 0.027 |
| | | | | | 8.504 |
| Num.Obs. | 3046 | 3046 | 3046 | 3046 | 3046 |
| Num.Imp. | 30 | 30 | 30 | 30 | 30 |
| R2 | 0.079 | 0.107 | 0.136 | 0.113 | 0.152 |
| R2 Adj. | 0.068 | 0.096 | 0.125 | 0.101 | 0.141 |

\* $p < 0.05$, \*\* $p < 0.01$, \*\*\* $p < 0.001$.

*Appendix A.3. Robustness Check for Structural Equation Model*

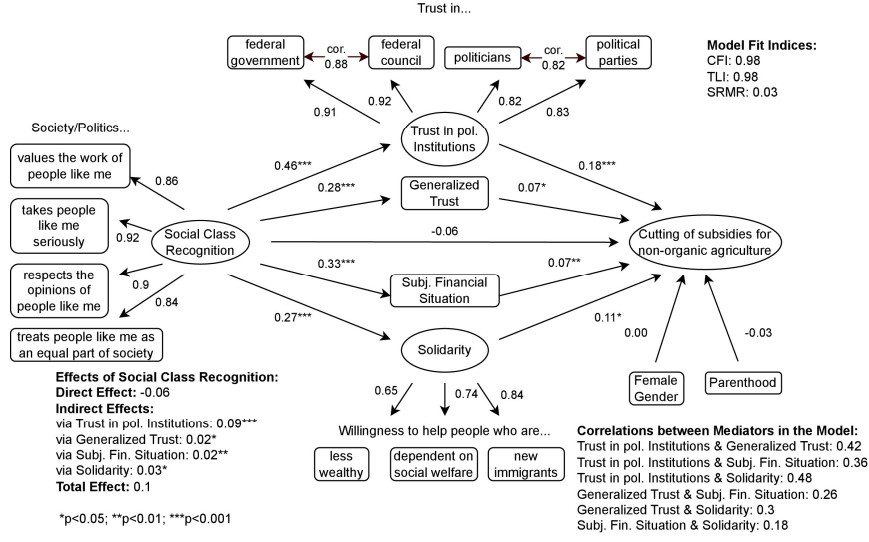

**Figure A1.** Cutting of Subsidies for Non-Organic Agriculture as Dependent Variable.

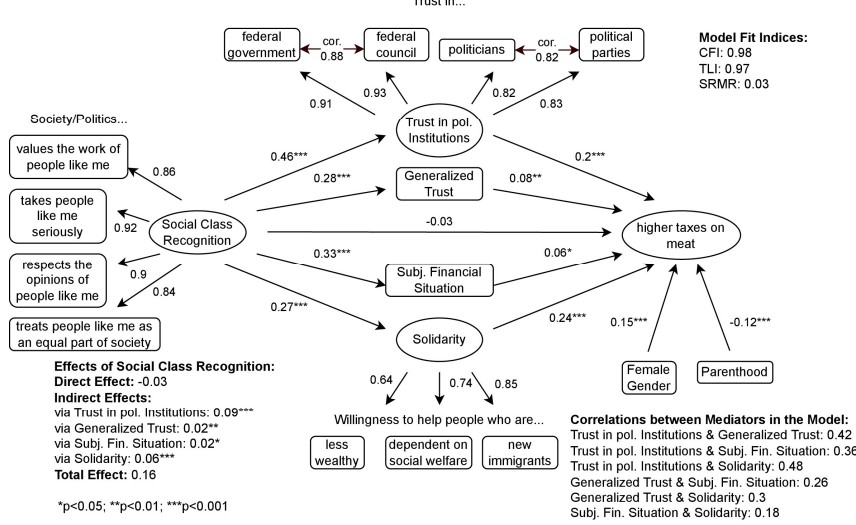

**Figure A2.** Higher Taxation of Meat as Dependent Variable.

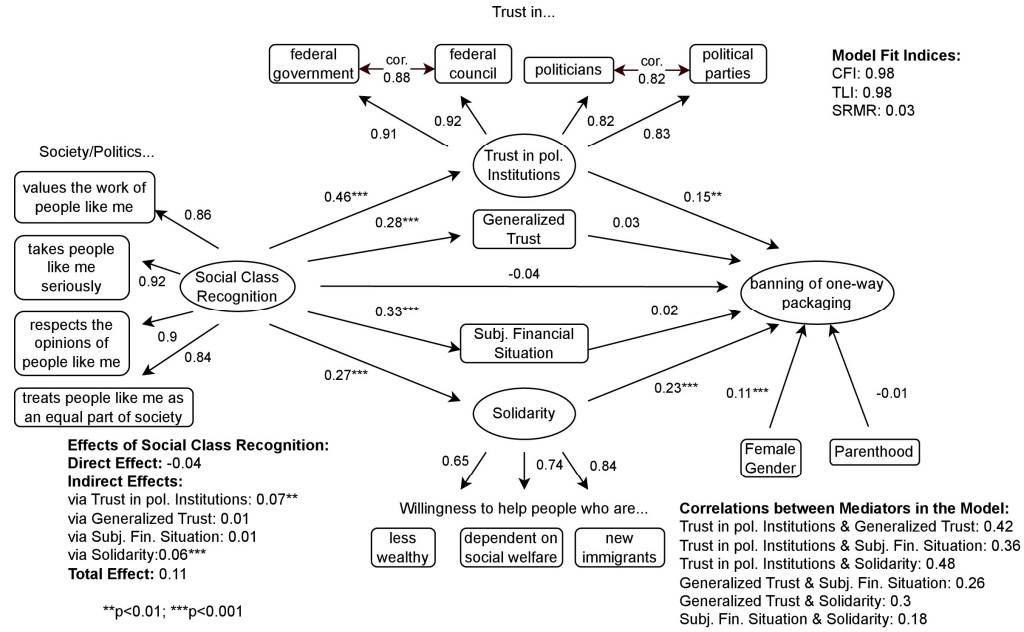

**Figure A3.** Banning of One-Way Packaging as Dependent Variable.

*Appendix A.4. Assumption Analyses of Structural Equation Model*

Appendix A.4.1. Mean Residuals, Standardized

**Table A9.** Mean standardized residuals indicating the mean difference between observed and estimated covariance matrix for each variable.

| Variable | S_iso_arb | S_iso_ernst | S_iso_mein | S_iso_wert | Ivpol_breg | Ivpol_btag | Ivpol_pol | Ivpol_part | Kids | Sex |
|---|---|---|---|---|---|---|---|---|---|---|
| Mean residual | 0.00 | 0.00 | 0.00 | 0.00 | −0.34 | −0.75 | −0.84 | −0.78 | 0.00 | 0.00 |
| Variable | Hilfeand_arm | | Hilfeand_soz | | HIlfeand_migr | Kwmass_agrar | Kwmass_fleisch | Kwmass_verpack | Finselbst | | V_vertr |
| Mean residual | 1.12 | | −1.21 | | −0.89 | −0.5 | 0.06 | 0.74 | 0.00 | | 0.00 |

Appendix A.4.2. Correlationmatrix of All Variables and Latent Constructs

**Table A10.** Correlation matrix of variables and latent constructs in the complete SEM, numbers in columns relate to variable names in lines.

| Variables | (1) | (2) | (3) | (4) | (5) | (6) | (7) | (8) | (9) | (10) | (11) | (12) |
|---|---|---|---|---|---|---|---|---|---|---|---|---|
| **(1) S_iso_arb** | 1.00 | | | | | | | | | | | |
| **(2) S_iso_ernst** | 0.79 | 1.00 | | | | | | | | | | |
| **(3) S_iso_mein** | 0.78 | 0.83 | 1.00 | | | | | | | | | |
| **(4) S_iso_wert** | 0.73 | 0.77 | 0.76 | 1.00 | | | | | | | | |
| **(5) Ivpol_breg** | 0.36 | 0.38 | 0.38 | 0.35 | 1.00 | | | | | | | |
| **(6) Ivpol_btag** | 0.37 | 0.39 | 0.38 | 0.36 | 0.88 | 1.00 | | | | | | |
| **(7) Ivpol_pol** | 0.33 | 0.35 | 0.34 | 0.32 | 0.75 | 0.76 | 1.00 | | | | | |
| **(8) Ivpol_part** | 0.33 | 0.35 | 0.34 | 0.32 | 0.75 | 0.76 | 0.82 | 1.00 | | | | |
| **(9) Hilfeand_arm** | 0.15 | 0.16 | 0.16 | 0.15 | 0.28 | 0.29 | 0.25 | 0.26 | 1.00 | | | |
| **(10) Hilfeand_soz** | 0.17 | 0.18 | 0.18 | 0.17 | 0.32 | 0.33 | 0.29 | 0.29 | 0.47 | 1.00 | | |
| **(11) Hilfeand_migr** | 0.20 | 0.21 | 0.21 | 0.19 | 0.37 | 0.38 | 0.34 | 0.34 | 0.55 | 0.63 | 1.00 | |
| **(12) Kwmass_agrar** | 0.10 | 0.10 | 0.10 | 0.10 | 0.23 | 0.23 | 0.20 | 0.21 | 0.16 | 0.18 | 0.21 | 1.00 |
| **(13) Kwmass_fleisch** | 0.14 | 0.15 | 0.15 | 0.14 | 0.33 | 0.34 | 0.30 | 0.30 | 0.23 | 0.27 | 0.31 | 0.52 |

**Table A10.** *Cont.*

| Variables | (13) | (14) | (15) | (16) | (17) | (18) | (19) | (20) | (21) | (22) | | |
|---|---|---|---|---|---|---|---|---|---|---|---|---|
| (14) Kwmass_verpack | 0.08 | 0.09 | 0.09 | 0.08 | 0.20 | 0.20 | 0.18 | 0.18 | 0.14 | 0.16 | 0.18 | 0.31 |
| (15) Finselbst | 0.28 | 0.30 | 0.29 | 0.28 | 0.33 | 0.34 | 0.30 | 0.30 | 0.11 | 0.13 | 0.15 | 0.13 |
| (16) V_vertr | 0.24 | 0.26 | 0.25 | 0.23 | 0.38 | 0.39 | 0.34 | 0.35 | 0.19 | 0.22 | 0.26 | 0.17 |
| (17) Kids | 0.00 | 0.00 | 0.00 | 0.00 | 0.00 | 0.00 | 0.00 | 0.00 | 0.00 | 0.00 | 0.00 | −0.06 |
| (18) D_sex | 0.00 | 0.00 | 0.00 | 0.00 | 0.00 | 0.00 | 0.00 | 0.00 | 0.00 | 0.00 | 0.00 | 0.08 |
| **Latent Constructs** | | | | | | | | | | | | |
| (19) S_iso | 0.86 | 0.92 | 0.90 | 0.84 | 0.42 | 0.43 | 0.38 | 0.38 | 0.17 | 0.20 | 0.23 | 0.11 |
| (20) Ivpol | 0.40 | 0.42 | 0.41 | 0.39 | 0.91 | 0.93 | 0.82 | 0.82 | 0.31 | 0.35 | 0.41 | 0.25 |
| (21) Hilfe_and | 0.23 | 0.25 | 0.24 | 0.23 | 0.44 | 0.45 | 0.39 | 0.40 | 0.64 | 0.74 | 0.85 | 0.25 |
| (22) Kwmass | 0.16 | 0.17 | 0.17 | 0.16 | 0.38 | 0.39 | 0.34 | 0.35 | 0.27 | 0.31 | 0.35 | 0.60 |

| Variables | (13) | (14) | (15) | (16) | (17) | (18) | (19) | (20) | (21) | (22) |
|---|---|---|---|---|---|---|---|---|---|---|
| (1) S_iso_arb | | | | | | | | | | |
| (2) S_iso_ernst | | | | | | | | | | |
| (3) S_iso_mein | | | | | | | | | | |
| (4) S_iso_wert | | | | | | | | | | |
| (5) Ivpol_berg | | | | | | | | | | |
| (6) Ivpol_btag | | | | | | | | | | |
| (7) Ivpol_pol | | | | | | | | | | |
| (8) Ivpol_part | | | | | | | | | | |
| (9) Hilfeand_arm | | | | | | | | | | |
| (10) Hilfeand_soz | | | | | | | | | | |
| (11) Hilfeand_migr | | | | | | | | | | |
| (12) Kwmass_agrar | | | | | | | | | | |
| (13) Kwmass_fleisch | 1.00 | | | | | | | | | |
| (14) Kwmass_verpack | 0.45 | 1.00 | | | | | | | | |
| (15) Finselbst | 0.19 | 0.11 | 1.00 | | | | | | | |
| (16) V_vertr | 0.25 | 0.15 | 0.26 | 1.00 | | | | | | |
| (17) Kids | −0.09 | −0.05 | 0.00 | 0.00 | 1.00 | | | | | |
| (18) D_sex | 0.12 | 0.07 | 0.00 | 0.00 | 0.06 | 1.00 | | | | |
| **Latent Constructs** | | | | | | | | | | |
| (19) S_iso | 0.17 | 0.10 | 0.33 | 0.28 | 0.00 | 0.00 | 1.00 | | | |
| (20) Ivpol | 0.37 | 0.22 | 0.36 | 0.42 | 0.00 | 0.00 | 0.46 | 1.00 | | |
| (21) Hilfe_and | 0.36 | 0.22 | 0.18 | 0.30 | 0.00 | 0.00 | 0.27 | 0.48 | 1.00 | |
| (22) Kwmass | 0.87 | 0.52 | 0.22 | 0.28 | −0.10 | 0.14 | 0.19 | 0.42 | 0.42 | 1.00 |

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
