# Peer review of "The Effect of Social Recognition on Support for Climate Change Mitigation Measures"

_sustainability, doi:10.3390/su152316486_

Round 1

Reviewer 1 Report

Comments and Suggestions for Authors

â–º Summary

The paper is well written and presents an interesting study on how “social recognition” affects the public support of climate change mitigation measures. The data used are very well presented and easy to read. The study only requires a few major and minor improvements before proceeding to publications. Those improvements mostly refer to improving the explanation of the methodological steps and materials and the contextual depth of the study.

â–º Major Comments

1.     Given the importance of the term “social recognition” for this study the manuscript lacks a clear definition of the term. In lines 143-144 the phrase “ The origins of recognition theory refer to recognition as a feeling which individuals 143 experience in intersubjective relations with relevant others” seems to address this but it is a bit perplexing that the term “recognition theory” is used rather than “social recognition”. Please clarify this definition (e.g. are those two terms considered the same by you?). Also, a simple definition of the term earlier in the text, in the introduction, would also help the reader of the study.

2.     On Section “3.1. Data”. I believe that there is an important bit of information lacking. How where the respondent of the online study “recruited” ? Where they voluntary participants? Where they part of an online survey platform? I think this is important information for the context of your study and needs to be described in detail.

3.     For the “Discussion” and “Introduction” section. This comment does not have to do with the quality of your own work, within the previous sections, but rather on the theoretical context within which it is placed. It is common and reasonable that scientists cannot be experts in all scientific areas and will need to acknowledge conclusions from other scientific areas in order to progress their work. For example, lines 535-538 present such a part of your work. I have to raise to your consideration that the contextual foundations stated by you in this part have received scientific criticism, which I believe should also be acknowledged, to some extent:

E.g. in relevance to “political change is slow and support for actual measures is disputed”

It can be argued that political change and support for measures is quite quick, changing the energy mix of Europe, with wind and solar energy, with double digits within 17 years:

https://ec.europa.eu/eurostat/statistics-explained/index.php?title=Renewable_energy_statistics#Share_of_renewable_energy_more_than_doubled_between_2004_and_2021

which have been associated with important impacts (that are also relevant to how climate change measures are perceived).

E.g.

Economic impacts:

Boccard, N. (2009). Capacity factor of wind power realized values vs. estimates. energy policy, 37(7), 2679-2688.

Questionable environmental performance/significance:

Wang, L., Wang, Y., Zhou, Z., Garvlehn, M. P., & Bi, F. (2018). Comparative Assessment of the Environmental Impacts of Hydro-Electric, Nuclear and Wind Power Plants in China: Life Cycle Considerations. Energy procedia, 152, 1009-1014.

Koutsoyiannis, D.; Onof, C.; Kundzewicz, Z.W.; Christofides, A. On Hens, Eggs, Temperatures and CO2: Causal Links in Earth’s Atmosphere. Sci 2023, 5, 35. https://doi.org/10.3390/sci5030035

Landscape impacts:

 Ioannidis, R.; Koutsoyiannis, D. A Review of Land Use, Visibility and Public Perception of Renewable Energy in the Context of Landscape Impact. Applied Energy 2020, 276, 115367, doi: https://doi.org/10.1016/j.apenergy.2020.115367.

among others.

4.      Need for separating the current “Discussion” section into a separate “Discussion” and “Conclusions section”. In its current state the “Discussion” section is actually more close to a “Conslusions” section. So I think you should put more effort to generating a new discussion section according to the guidelines of the journal: “Authors should discuss the results and how they can be interpreted in perspective of previous studies and of the working hypotheses. The findings and their implications should be discussed in the broadest context possible and limitations of the work highlighted. Future research directions may also be mentioned. This section may be combined with Results”. You have a lot of literature already within the text so you can discuss your findings within the context of this literature.

â–º Minor Comments

i.       Not all people reading your article will have knowledge to what age groups the generations referred to in Figure 2 consist of. To help the readers of the stud, since those generation groups are currently not analyzed within the text I think you should refer more explicitly to which particular age groups they refer to whether within the Figure or within the text (maybe within existing lines 377-380).

ii.     The comments I made in major comment no 3. are also relevant to the measures that you examine as relevant to climate change mitigation, as described in lines 334-336 and in the Appendix (called “climate measures”). Are those indeed measures that are pivotal to the environmental well-being of the planet? Are there any different scientific opinions in this matter? Since I am not an expert on this field, I do not insist on including such considerations but leave it to your judgement.

iii.   Points like the one mentioned in lines 219-229 seem very reasonable and could maybe help you in discussing the implications of your results in the new Discussion section, in a broader political and social context.

Reviewer 2 Report

Comments and Suggestions for Authors

The topic of the paper is very interesting and my minor suggestion for this paper. Authors the hypotheses should present in the literature review, not in separate paragraph. I would be easier to follow the background of the model. Furthermore, authors schematically should present this model. It is not clear how authors constructed the scales, where were the statistical analysis of that, what literature they referred to. Where are the assumtion analysis of SEM model? 

Reviewer 3 Report

Comments and Suggestions for Authors

Dear,

I think this is an interesting manuscript that discuss and present results which are important in a time with difficulties to handle changes in climate. Unfortunately I think the results are not very good presented due to figures that are not easy to understand. The figures must be improved. The paper would also be stronger if there could be more references with similar studies in other countries. Not only German categories but also other if possible.

Comments related to lines in the manuscript:

Line 60 - Do we need examples that groups are affected differently?

Line 97 - Is the word dangerous correct to use? If it is used I think there is a need to describe this more.

Line 280 - Hypotheses 2 - How is this when it is more dangeorus for  younger generation? Is this hypotheses correct due to earlier description of young and older generations?

Line 310 - Online survey. Describe more where you did this online survey, Google ?, in what region etc? Time schedule for answers and how many did you try to reach, % of answers? There is a need for more data that decribes data collection.

Line 353 - Discuss if Germany is different form other countries.

Line 378 - You have to clarify the different groups. In the figure you use abbreviations which are not described earlier.

Line 404 - Text should be added before the figures! More detailed description of the data in the legends of the figure. Category name in the text and what does positive values indicate? The figures should be in shape that it could be understood without reading the text.

Line 420 - The same as for figure 1 must be corrected for figure 2 and the abbreviations are not explained. What does higher value of social recognition mean?

Line 455 - Text before fig 3 and also here there is a need för a better description of the results in the figure. Positive value indicate smallest influence? (in the text) Explain!

Line 469 - Figure 4. A need for better descripiton in the legend of the figure. What does brackets with r, s, t and m indicate? This is explained in the appendix but must be shown in relation to the figure.

Line 534 - The discussion is rather short and it would have been nice if you could relate to other similar studies. Are there any differences bewteen group from west and east region in Germany? I think the paper would be stronger if you try to discuss weak and strong points in your study and how to further develope a new research study like this.

Round 2

Reviewer 1 Report

Comments and Suggestions for Authors

Dear Authors

Thank you for addressing the comments in detail. I provide some last minor comments before the publication.

B.1) Regarding my original comment no 2. I believe the description of the selection of the group is satisfactory now. One last note has to do with the phrase “After checking for dubious response behavior 78 further respondents had to be excluded …” . Even though 78 is a small number and doesn’t seem to affect your results (given the 3124 participants) I believe for scientific clarity you need to report what was the dubius response behaviour exactly and how you identified it.

B.2) Regarding your response to my original comment no 3. First of all, thank you too for your detailed response. I believe that the addition of the new text makes the scientific background clearer. However, I still suggest that a scientific study should acknowledge different contemporary perspectives/constructive criticism of existing practices-methods, even when it does not fundamentally agree with them. I believe that in a study including more than 100 citations this is quite reasonable. I note that the IPPC, which is the primary institution on whose scientific results you base your work, embraces constructive criticism. E.g. one of the studies I provided is co-authored by a collective nobel price winner of the IPCC and another is cited multiple times within IPPC’s reports chapter 12. I do not insist on the above but leave it to your judgement. Also the citations I provided are only indicative and in no case are they the only relevant ones or the most important ones.

B.3) Regarding the text additions in relation to my minor comment no ii). For the justification of the selected measures your note in lines 358-359 that “… they relate to everyday consumption which has been identified as one of the main drivers of global warming [89].”. I believe that in an a time in which are seeing again a rise in wars globally (e.g. Ukraine that you mention of Israel) we should also acknowledge the importance of other important sources of emissions that are affected by them and will be affected even further due to potential escalations. E.g. (i) industrial emissions (including from the military industry and others) and (ii) manufacturing and constructions emissions (including for rebuilding destroyed areas). Those two add up to approx. 8 billion t per year (https://ourworldindata.org/emissions-by-sector) and I believe that should also be referred to when discussing adaptation policies and social support thereof.

Reviewer 2 Report

Comments and Suggestions for Authors

-

Author Response

I understand you are content with my changes. Thank you for your previous remarks!

Reviewer 3 Report

Comments and Suggestions for Authors

I thank the authors to the good revision of the manuscript they have made. However they should add a reference to Figur 2 in the text before the figure.
